# Scalable Intervention Target Estimation in Linear Models

**Burak Varıcı**
Rensselaer Polytechnic Institute
varicb@rpi.edu

**Karthikeyan Shanmugam**
IBM Research AI
karthikeyan.shanmugam2@ibm.com

**Prasanna Sattigeri**
IBM Research AI
psattig@us.ibm.com

**Ali Tajer**
Rensselaer Polytechnic Institute
tajer@ecse.rpi.edu

## Abstract

This paper considers the problem of estimating the unknown intervention targets in a causal directed acyclic graph from observational and interventional data. The focus is on soft interventions in linear structural equation models (SEMs). Current approaches to causal structure learning either work with known intervention targets or use hypothesis testing to discover the unknown intervention targets even for linear SEMs. This severely limits their scalability and sample complexity. This paper proposes a scalable and efficient algorithm that consistently identifies all intervention targets. The pivotal idea is to estimate the intervention sites from the difference between the precision matrices associated with the observational and interventional datasets. It involves repeatedly estimating such sites in different subsets of variables. The proposed algorithm can be used to also update a given observational Markov equivalence class into the interventional Markov equivalence class. Consistency, Markov equivalence, and sample complexity are established analytically. Finally, simulation results on both real and synthetic data demonstrate the gains of the proposed approach for scalable causal structure recovery. Implementation of the algorithm and the code to reproduce the simulation results are available at https://github.com/bvarici/intervention-estimation.

## 1 Introduction

Directed acyclic graphs (DAG) are commonly used for encoding the cause-effect relationships among random variables. Extensive research has been dedicated to learning the structure of DAGs from their associated observational data. Structure learning from the observational data relies on uncovering conditional independence (CI) among the random variables. Since structurally distinct DAGs can encode the same set of CI relations, a DAG is identifiable only up to its Markov equivalence class (MEC) from the observational data. Subsequently, *interventional* data can be used to further refine the MEC obtained from the observational data and learn specific causal effects.

This paper is motivated by addressing two significant independent challenges in causal discovery. First, most of the existing approaches for learning with interventional datasets require the *intervention target set* to be known, which can be a strong assumption. For instance, gene-editing technologies are known to perform cleavage at off-target genome sites [1]. Therefore, identifying the *intervened* nodes alone is a critical problem in structure learning, and despite its significance, it remains uninvestigated. Secondly, besides learning the structures of single DAGs, there exist application domains in which the goal is learning the structural changes between two related networks and their associated DAGs, or learning the sites of interventions. For instance, structural differences between the gene regulatory

35th Conference on Neural Information Processing Systems (NeurIPS 2021).

networks of different subtypes of cancers can help to identify the roles of specific genes [2]. In electroencephalography analysis, the objective is to detect different brain regions that have different interactions when the subject is performing various tasks [3]. These brain regions correspond to intervened nodes in a causal graph representation. Another application area is fault detection in large-scale Internet of things and cloud applications [4]. Faulty nodes in the system can be considered as intervened nodes, and they are localized through intervention target estimation.

In practice, fixing the target variable at a specific value or removing its causal dependencies is often difficult, while disturbing the distribution of a target variable is easier [5]. The type of interventions that do not remove the causal effects are commonly observed in the real world. For example, elements of an advertising system can be modified without removing the causal effects [6]. In another example, consider molecular biology, in which the effects of infused chemicals to the cell are not set to specific values nor are they known precisely [7,8]. Therefore, we consider a *soft intervention* setting, in which we assume the conditional distributions of the target variables are changed, but no assumption is made on the causal effects. Finally, we assume that the topological ordering remains the same after the intervention procedure.

Under the soft intervention model, we propose an algorithm for estimating the intervention targets given the data from two linear SEMs associated with the observational and interventional data. Motivated by the fact that the difference of the precision matrices associated with these two models is sparse, we focus on estimating the sparse difference between precision matrices to avoid extensive conditional independence testing. This leads to a significant improvement in the computational complexity compared to those of the alternative methods. This facilitates scaling up to high-dimensional settings. Furthermore, we show that this algorithm can be used in conjunction with an observational DAG learning procedure to refine the MEC to *interventional*-MEC ($\mathcal{I}$-MEC). Besides being consistent in the population setting, we provide the finite-sample guarantees for linear SEM with Gaussian noise when the soft interventional changes between the two models are sparse. Our main contributions are as follows:

- We propose an algorithm that identifies intervention targets under the intervention-faithfulness assumption. We show that our algorithm identifies $\mathcal{I}$-MEC given the observational MEC.

- We provide the sample complexity of our algorithm under linear SEM with Gaussian noise.

- We perform experiments on both real biological and synthetic datasets to illustrate the ability to work in the high-dimensional settings and the gains compared to the relevant methods.

## 2   Related work

Among the broad range of approaches to intervention recovery, there exist two methods that are closely related to the scope of this paper: (i) estimating the difference between two DAGs, and (ii) learning from a combination of observational and interventional datasets.

**Direct estimation of differences:** Direct estimation of differences in linear SEMs has been studied recently. The study in [9] proposes a PC-style algorithm for learning changes in the edge weights by testing invariances of regression coefficients and noise variances. Even though the differences can be sparse, individual models can be dense, and estimating these variables through regression can be inaccurate. Furthermore, the number of hypothesis tests is exponential in the number of nodes that are affected by the changes, which can be prohibitive even under sparse changes in the hub nodes. Estimating the difference of two precision matrices is a relatively easier task and has received attention recently [10–13], providing finite-sample guarantees in the high-dimensional regime when such a difference is sparse. An existing study closest to the scope of our work is [14], which proposes re-estimating precision difference to progressively eliminate nodes and estimate the difference DAG. This approach critically hinges on the assumption that the noise variance is invariant, rendering limited applicability to intervention settings. In contrast, our algorithm builds on the changes in noise variances and estimates the intervention targets efficiently. We demonstrate the effect of this difference in Appendix B.1.

**Learning interventional-MEC:** There is a growing number of studies on causal structure learning from both observational and interventional data. Score-based greedy interventional equivalence search (GIES) [15] and hybrid interventional greedy sparsest permutation (IGSP) [16] algorithms are proposed for settings in which there are no latent confounders. Both of these algorithms assume

that the intervention targets are known. However, knowing the intervention targets can be a strong assumption since even for controlled interventional experiments, off-target effects are common. For instance, noise variances of the off-target variables can change, resulting in the intervened nodes being unknown.

For causal structure learning without the knowledge of the intervention targets, an existing study includes the dynamic programming approach in [8]. This approach has limited scalability due to its time complexity being exponential in the model size. Structural discovery from interventions proposed in [17] is a neural network-based method that can learn from interventional data without target knowledge. Nevertheless, it requires discrete data and can have at most one intervened node in a given setting, limiting its applicability. The differential causal discovery from interventional data (DCDI) algorithm proposed in [18] extends the approach of [17] without making as strong assumptions, e.g., learning a distribution over all potential interventional families via continuous optimization. While this method is shown to converge to the real intervention targets, its runtime becomes prohibiting even in models with as few as 100 nodes. The Unknown-target IGSP (UT-IGSP) algorithm proposed in [19] learns the intervention targets simultaneously while learning the causal structure. Even though interventions refine the search space, a greedy search of the sparsest permutation is still too slow in the high-dimensional regime, especially when using non-Gaussian CI tests. A graphical characterization of soft interventions with unknown targets is proposed in [7] for causally insufficient systems. This algorithm, however, relies on CI tests and it is not scalable. We focus on causally sufficient systems in this paper.

## 3 Problem definition

Let $\mathcal{G} \triangleq ([p], E)$ be a DAG with the node set $[p] \triangleq \{1, \ldots, p\}$ and the edge set $E \subseteq [p] \times [p]$. We denote the directed edge from $i \in [p]$ to $j \in [p]$ by $i \to j$. We associate a random variable $X_i$ with $i \in [p]$, and accordingly, define the random vector $X \triangleq (X_1, \ldots, X_p)^\top$. We consider a linear SEM, according to which

$$X = B^\top X + \epsilon \,, \tag{1}$$

where $B \in \mathbb{R}^{p \times p}$ is the autoregressive matrix in which $B_{ij} \neq 0$ if and only if $i \to j$ in $\mathcal{G}$. The random vector $\epsilon \in \mathbb{R}^{p \times 1}$ has zero-mean and covariance matrix $\Omega \triangleq \mathsf{diag}(\sigma_1^2, \ldots, \sigma_p^2)$. We denote the covariance matrix of $X$ by $\Sigma$ and its inverse (the precision matrix) by $\Theta$, which satisfies $\Theta = (I - B)\Omega^{-1}(I - B)^\top$. For entries of $\Theta$, we have

$$\Theta_{ij} = -\frac{B_{ij}}{\sigma_j^2} - \frac{B_{ji}}{\sigma_i^2} + \sum_{k \in \mathsf{ch}(i) \cap \mathsf{ch}(j)} \frac{B_{ik} B_{jk}}{\sigma_k^2} \,, \quad \forall i \neq j \,, \tag{2}$$

$$\text{and} \quad \Theta_{ii} = \sigma_i^{-2} + \sum_{j \in \mathsf{ch}(i)} \sigma_j^{-2} B_{ij}^2 \,, \quad \forall i \in [p] \,, \tag{3}$$

where $\mathsf{ch}(i), \mathsf{pa}(i), \mathsf{de}(i),$ and $\mathsf{an}(i)$ denote children, parents, descendants, and ancestors set of node $i$ in DAG $\mathcal{G}$, respectively.

From the observational data, a DAG can be learned only up to its MEC [20]. Interventions are used to increase the identifiability of a DAG by removing all causes of the intervention target (*perfect intervention*) or modifying those relationships without removing them completely (*imperfect intervention*). We consider the following *soft intervention* setting, which does not remove causal effects (from direct parents) on intervention target nodes, and hence, is more practical.

**Soft intervention model.** In this model, interventions correspond to disturbing the target nodes $i \in \mathcal{I}$ by changing the variances of their noise variables, while the cause weights, i.e., the weights $B_{\mathsf{pa}(i),i} \triangleq \{B_{j,i} : j \in \mathsf{pa}(i)\}$, can vary freely. This intervention procedure on the initial DAG results in a second DAG with new parameters.

Let $\mathcal{G}^{(1)}$ represent a linear SEM prior to intervention with parameters $B^{(1)}, \epsilon^{(1)}$, and $\mathcal{G}^{(2)}$ be the linear SEM after the intervention, with parameters $B^{(2)}, \epsilon^{(2)}$. The intervention target set that relates these two DAGs is

$$\mathcal{I} \triangleq \{i : \sigma_i^{(1)} \neq \sigma_i^{(2)}\} \,. \tag{4}$$

Accordingly, denote the covariance and precision matrices of these two models by $\Sigma^{(1)}, \Sigma^{(2)}, \Theta^{(1)}$, and $\Theta^{(2)}$. Accordingly, denote the differences between the two models by $\Delta_B \triangleq B^{(1)} - B^{(2)}$ and $\Delta_\Theta \triangleq \Theta^{(1)} - \Theta^{(2)}$. For a subset of nodes $S \subseteq [p]$, denote the precision matrix of the random vector $X_S \triangleq \{X_i : i \in S\}$ by $\Theta_S$. We also denote the set of changed nodes by $S_\Delta \triangleq \{k : (\Delta_\Theta)_{k,k} \neq 0\}$ and denote its size by $p_\Delta \triangleq |S_\Delta|$. According to (3), this set consists of all the intervened nodes and their parents.

In this paper, we estimate the intervention targets $\mathcal{I}$ given the data from two SEMs under the soft intervention model. Furthermore, we estimate the non-intervened parents of the targets, which update the given observational MEC into the $\mathcal{I}$-MEC. Formally, we define $\psi : \Sigma^{(1)} \times \Sigma^{(2)} \rightarrow \left( \hat{\mathcal{I}}, \mathsf{pa}(\hat{\mathcal{I}}) \right)$ as the estimator that maps the covariance matrices of the observational and interventional data to an intervention target set estimate and their parents. We aim to maximize the probabilities of $\psi$ recovers $\mathcal{I}$ and their parents. To this end, we define

$$\mathsf{P} \triangleq \mathbb{P}(\mathcal{I} = \hat{\mathcal{I}}) \qquad \text{and} \qquad \mathsf{Q} \triangleq \mathbb{P}(\mathsf{pa}(\mathcal{I}) = \mathsf{pa}(\hat{\mathcal{I}})) . \qquad (5)$$

We will show that the algorithm proposed in Section 4 exactly recovers $\mathcal{I}$ and the non-intervened parents of the members of $\mathcal{I}$. Hence, given the observational MEC, we obtain the $\mathcal{I}$-MEC.

## 4 Algorithm and main results

In this section, we provide our proposed algorithm and present the attendant performance guarantees. Our algorithm involves repeatedly estimating the difference of the precision matrices to find the intervention target set $\mathcal{I}$, or equivalently, its complement $\mathcal{I}^{\mathrm{C}}$. This algorithm consists of three key steps. In **Step 1**, instead of directly estimating $\mathcal{I}$, we aim to discard the nodes that are strongly deemed not to belong to $\mathcal{I}$. For this purpose, we start by identifying the non-intervened nodes that do not have intervened children. These nodes are not of interest and they are discarded from further consideration. We achieve this by estimating difference of precision matrices corresponding to the complete model with variables $[p]$, and we denote the changed nodes in the diagonal of this precision difference matrix with $S_\Delta$. We continue with only the nodes contained in $S_\Delta$ for further scrutiny. A naive approach to identifying the rest of the non-intervened nodes is computing $\Delta_\Theta$ exhaustively for all the $2^{p_\Delta}$ subsets of $S_\Delta$, which for large $p_\Delta$ is computationally prohibitive. Alternatively, we partition $S_\Delta$ into two sets: the set of *non-intervened source* nodes and its complement. We feed these two partitions for further processing to Step 2. Since the distribution of a node relies only on its ancestors, reaching a topological ordering is critical to reduce the complexity. In **Step 2**, for each source node, we find the nodes that share a common ancestor with it. Subsequently, we decompose $S_\Delta$ into equivalence classes according to these ancestral relationships with non-intervened source nodes. Decomposing $S_\Delta$ allows us to order the equivalence classes according to a topological ordering. In **Step 3**, we process these classes individually, and show that we only need to compute $\Delta_\Theta$ for all the subsets of each class considered in Step 3. This results in a significant reduction in the computational complexity compared to the exhaustive search approach. Finally, we identify the non-intervened parents of $\mathcal{I}$ from the earlier results. Since estimating the precision difference appears in our algorithm repeatedly, we describe it next and then continue with the details of the three steps.

**Precision difference estimation (PDE).** When the difference between two SEMs is sparse, estimating the difference of their precision matrices can be formulated as a Lasso-type problem and solved efficiently. Since it will be used repeatedly, it is important for this function to recover the support of $\Delta_\Theta$ and have a feasible computational complexity. The study in [12] solves the following convex problem through the alternating direction method of multipliers (ADMM) to estimate $\Delta_\Theta = \Theta^{(1)} - \Theta^{(2)}$:

$$\hat{\Delta}_\Theta = \operatorname*{argmin}_{\Delta_\Theta} \left\{ \frac{1}{2} \mathsf{Tr}(\Delta_\Theta^\top \hat{\Sigma}^{(1)} \Delta_\Theta \hat{\Sigma}^{(2)}) - \mathsf{Tr}(\Delta_\Theta(\hat{\Sigma}^{(1)} - \hat{\Sigma}^{(2)})) + \lambda \|\Delta_\Theta\|_1 \right\} , \qquad (6)$$

where $\lambda$ is a tuning parameter. The computational complexity of this algorithm grows according to $O(p^3)$, which overcomes the limitation of the algorithm of [10] used in [14], which has complexity $O(p^4)$. Therefore, we use this ADMM-based method as our PDE procedure.

**Step 1: Finding the non-intervened source nodes.** By leveraging the PDE procedure discussed, we first estimate $\Delta_\Theta$ over all $[p]$ nodes and obtain $S_\Delta$. Representation of $\Theta_{i,i}$ in (3) implies that

a diagonal entry of $\Delta_\Theta$ is non-zero if and only if either its corresponding node is in $\mathcal{I}$ or it has a child in $\mathcal{I}$. Hence, the set $[p] \setminus S_\Delta$ contains only non-intervened nodes and can be discarded from consideration. Furthermore, some non-intervened nodes in $S_\Delta$, namely those that do not have intervened ancestors, do not observe a change in their statistics. Therefore, it is possible to identify them from covariance matrices $\Sigma^{(1)}$ and $\Sigma^{(2)}$. Subsequently, these nodes can serve as the starting points for distinguishing the rest of the $\mathcal{I}^{\mathrm{C}}$ in $S_\Delta$. Let us define them as the *non-intervened source nodes*, denoted by

$$J_0 \triangleq \{j : j \in S_\Delta,\ j \notin \mathcal{I},\ \mathsf{an}_{\mathcal{I}}(j) = \emptyset\}\ . \tag{7}$$

The outputs of this step, $S_\Delta$ and $J_0$, are fed into the next steps of the algorithm.

**Step 2: Forming equivalence classes from $J_0$.** We will show that for any non-intervened node $j \in S_\Delta$, there exists a minimal subset of $S_\Delta$, which makes the corresponding diagonal entry of the precision matrix invariant, and it does not contain any descendant of $j$. Therefore, the non-intervened nodes that have the same ancestors are affected by the interventions similarly, and finding their ancestors is critical. We show that determining whether node $k \in S_\Delta \setminus J_0$ has a common ancestor with node $j \in J_0$ is possible by applying PDE on $\{j, k\}$. Accordingly, we define the *source ancestral set $J_0^k$* for each node $k \in S_\Delta \setminus J_0$ as

$$J_0^k \triangleq \{j : j \in J_0,\ \mathsf{an}(j) \cap \mathsf{an}(k) \neq \emptyset\}, \quad \forall k \in S_\Delta \setminus J_0\ . \tag{8}$$

Next, we decompose the set $S_\Delta \setminus J_0$ into *equivalence classes* where all the nodes in a class have the same source ancestral set. We denote these equivalence classes by $\mathcal{A}_1, \ldots, \mathcal{A}_L$, and the source ancestral set corresponding to a class $\mathcal{A}_\ell$ by $J_0^{\mathcal{A}_\ell}$ for $\ell \in [L]$. These classes are ordered according to a topological order such that for $1 \leq \ell < \ell' \leq L$, $J_0^{\mathcal{A}_{\ell'}} \not\subset J_0^{\mathcal{A}_\ell}$. In other words, the class corresponding to the superset of any $J_0^{\mathcal{A}_\ell}$ should appear later than $\mathcal{A}_\ell$ in the sequence $\mathcal{A}_1, \ldots, \mathcal{A}_L$. Source ancestral sets and equivalence classes are fed into the next step.

**Step 3: Processing equivalence classes.** Given Step 1 and Step 2, we can describe our exact search space of subsets for $\Delta_\Theta$ estimates to declare whether a node is intervened. We process equivalence classes $\mathcal{A}_1, \ldots, \mathcal{A}_L$ individually, i.e., at stage $\ell$, we consider the nodes in $\mathcal{A}_\ell$. Let us define $\mathcal{M}_\ell \triangleq J_0 \cup \bigcup_{1 \leq b < \ell} \mathcal{A}_b$. It suffices to estimate $\Delta_{\Theta_{\mathcal{M}_\ell \cup A}}$ **only for each** $A \subseteq \mathcal{A}_\ell$ to determine the intervention status of any node in $\mathcal{A}_\ell$ class. This key observation reduces the number of PDE steps needed. Specifically, for any non-intervened $j \in \mathcal{A}_\ell$, there exists a subset $A \subseteq \mathcal{A}_\ell$ such that the corresponding diagonal entry of $\Delta_{\Theta_{\mathcal{M}_\ell \cup A}}$ will be zero, and there does not exist any such set for the intervened nodes in $\mathcal{A}_\ell$. Formally, the *process equivalence class* returns

$$\mathcal{I}_\ell = \{i : i \in \mathcal{A}_\ell \cap \mathcal{I}\}, \quad \text{and} \quad J_\ell = \{j : j \in \mathcal{A}_\ell \cap \mathcal{I}^{\mathrm{C}}\}\ . \tag{9}$$

Finally, we identify the non-intervened parents of the intervened nodes without any new $\Delta_\Theta$ estimates.

**Computational complexity.** Algorithm 1 repeatedly performs PDE in each step. The number of required instances of PDE is $(p_\Delta + 1)$ in Step 1, $O(p_\Delta^2)$ in Step 2, and $\sum_{\ell \in [L]} 2^{|\mathcal{A}_\ell|}$ in Step 3. Hence, it grows exponentially with $\max_{\ell \in [L]} |\mathcal{A}_\ell|$, which can be $p_\Delta$ in the worst case in extreme examples. Nevertheless, in almost all practical scenarios it is usually considerably smaller. To provide some insights, we provide the next example and relegate more discussions to Appendix B.4.

**Example 1.** Consider a DAG with nodes $\{1, 2, 3, 4, 5\}$ and the edge set $\{1 \rightarrow 3, 3 \rightarrow 4, 2 \rightarrow 4, 2 \rightarrow 5, 4 \rightarrow 5\}$ and let $\mathcal{I} = \{3, 5\}$. Hence, we have $S_\Delta = \{1, 2, 3, 4, 5\}, J_0 = \{1, 2\}, J_0^3 = \{1\}, J_0^4 = \{1, 2\}, J_0^5 = \{1, 2\}$, and accordingly, $\mathcal{A}_1 = \{3\}, \mathcal{A}_2 = \{4, 5\}, J_0^{\mathcal{A}_1} = \{1\}$, and $J_0^{\mathcal{A}_2} = \{1, 2\}$. Note that the largest $\mathcal{A}$ class has 2 nodes whereas $S_\Delta$ has 5 nodes.

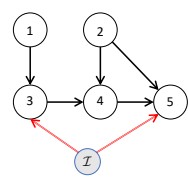

**Restricted SEM.** For a linear SEM $\mathcal{G}$ with $(B, \epsilon)$, we denote the restricted SEM (see Lemma 1 for details) that characterizes the relationship among the random variables $X_S$ for a set $S$ by $(B_S, \epsilon_S)$. As defined earlier, the corresponding precision matrix is denoted by $\Theta_S$. The entries of $B_S$ and noise variances $\sigma_S$ are characterized by the original values of $B$, $\sigma$, and $\Theta$.

**Remark 1** *We remark that the invariance of the distributions for the noise term of a node and the value of the node are equivalent only for the non-intervened nodes that do not have an intervened ancestor. Therefore, only such nodes can be detected from the full linear SEM. The noise term of a*

---
**Algorithm 1** Causal Intervention Target Estimator (CITE)

---
1: **Input:** $\Sigma^{(1)}$ and $\Sigma^{(2)}$
2: **Output.** $\mathcal{I}$ intervention set
3: Estimate $\Delta_\Theta \leftarrow$ *precision difference*$(\Sigma^{(1)}, \Sigma^{(2)})$
4: Form $S_\Delta \triangleq \{k : k \in [p], [\Delta_\Theta]_{k,k} \neq 0\}$
5: Form $J_0$ using (7) and $J_0^k$ for each $k \in S_\Delta \setminus J_0$ using (8)
6: Form equivalence classes $\mathcal{A}_1, \ldots, \mathcal{A}_L$.
7: **for** $\ell \in [L]$ **do**
8:     Take $\mathcal{A}_\ell$ set and the corresponding $J_0^{\mathcal{A}_\ell}$ set
9:     $\mathcal{B}_\ell \leftarrow \{b : J_0^{\mathcal{A}_b} \subset J_0^{\mathcal{A}_\ell}, \ 1 \leq b < \ell\}$
10:     $\mathcal{M}_\ell = J_0^{\mathcal{A}_\ell} \cup \bigcup_{b \in \mathcal{B}_\ell} \mathcal{A}_b$
11:     $J_\ell, \mathcal{I}_\ell \leftarrow$ *process equivalence class*$(\mathcal{M}_\ell, \mathcal{A}_\ell, \Sigma^{(1)}, \Sigma^{(2)})$
12: **end for**
13: $\mathcal{I} = \bigcup_{\ell \in [L]} \mathcal{I}_\ell$
14: $\text{pa}(\mathcal{I}) \leftarrow$ *parent finder*$(\mathcal{I}, \mathcal{J}, \mathcal{M}_1, \mathcal{A}_1, \ldots, \mathcal{M}_L, \mathcal{A}_L, \Sigma^{(1)}, \Sigma^{(2)})$
15: **Return** $\mathcal{I}$ and $\text{pa}(\mathcal{I})$

---

---
**Algorithm 2** Functions for the main algorithm

---
**Precision Difference Estimation (PDE) ($\Sigma^{(1)}, \Sigma^{(2)}$)**

---
1: Using ADMM based algorithm of [12] method estimate $\Delta_\Theta = (\Sigma^{(1)})^{-1} - (\Sigma^{(2)})^{-1}$
2: Symmetrize $\Delta_\Theta$: set $\Delta_\Theta = (\Delta_\Theta + \Delta_\Theta^\top)/2$
3: Threshold $\Delta_\Theta$: set $[\Delta_\Theta]_{i,j} = 0$ if $|[\Delta_\Theta]_{i,j}| < \varepsilon$.
4: **Return** $\Delta_\Theta$

---
**Process Equivalence Class ($\mathcal{M}, \mathcal{A}, \Sigma^{(1)}, \ \Sigma^{(2)}$)**

---
1: For each subset $A \subseteq \mathcal{A}$, estimate $\Delta_{\Theta_{\mathcal{M} \cup A}} \leftarrow$ *precision difference*$((\Sigma^{(1)})_{A,A}, (\Sigma^{(2)})_{A,A})$
2: **for** $k \in \mathcal{A}$ **do**
3:     **if** $\exists A \subseteq \mathcal{A}$, where $k \in A$, and $[\Delta_{\Theta_{\mathcal{M} \cup A}}]_{k,k} = 0$ **then**
4:         $J \leftarrow J \cup k$
5:     **else**
6:         $\mathcal{I} \leftarrow \mathcal{I} \cup k$
7:     **end if**
8: **end for**
9: **Return** $J, \mathcal{I}$

---
**Parent Finder ($\mathcal{I}, \mathcal{J}, \mathcal{M}_1, \mathcal{A}_1, \ldots, \mathcal{M}_L, \mathcal{A}_L, \Sigma^{(1)}, \Sigma^{(2)}$)**

---
1: **for** $i \in \mathcal{I}$ **do**
2:     $c_i \leftarrow c_i \triangleq \ell : i \in \mathcal{A}_\ell$
3:     **for** $j \in \mathcal{M}_{c_i}$ **do**
4:         **if** $\nexists A \subseteq \mathcal{A}_{c_i}$ such that $[\Delta_{\Theta_{\mathcal{M}_{c_i} \cup A}}]_{j,i} = 0$ **then**
5:             Add $j$ to $\text{pa}(i)$
6:         **end if**
7:     **end for**
8: **end for**
9: **Return** $\text{pa}(i)$ for $i \in \mathcal{I}$

---

*non-intervened node maintains its invariance in a restricted SEM in which we keep its intervened ancestors and their parents. On the other hand, the noise term of an intervened node is always variant for any choice of restricted SEM.*

In the following subsections, we will provide different analytical guarantees of Algorithm 1. Specifically, we will comment on the consistency of $\mathcal{I}$ recovery, the refinement of MEC to $\mathcal{I}$-MEC, and the sample complexity.

### 4.1 Consistency of $\mathcal{I}$ recovery

We provide the consistency of Algorithm 1 for estimating $\mathcal{I}$ in this subsection. First, we need the following assumption to ensure that interventions are successful.

**Assumption 1 ($\mathcal{I}$-faithfulness)** *For any choice of $i, j \in S \subseteq [p]$, we have the following properties:*

1. *If $\sigma_i^{(1)} \neq \sigma_i^{(2)}$, then $\sigma_{S,i}^{(1)} \neq \sigma_{S,i}^{(2)}$.*

2. *If $\sigma_{S,i}^{(1)} \neq \sigma_{S,i}^{(2)}$, then $[\Theta_S^{(1)}]_{i,i} \neq [\Theta_S^{(2)}]_{i,i}$.*

3. *If $\sigma_{S,i}^{(1)} \neq \sigma_{S,i}^{(2)}$ and $[B_S]_{j,i} \neq 0$ in either model, then $[\Theta_S^{(1)}]_{i,j} \neq [\Theta_S^{(2)}]_{i,j}$.*

Next, we characterize the parameters of a restricted SEM, and then formalize the observations stated in Remark 1 in the subsequent proposition.

**Lemma 1 ( [14])** *Corresponding to a subset $S \subseteq [p]$, denote the removed set of nodes by $U \triangleq [p] \setminus S$ and define $U_j \triangleq U \cap \mathsf{an}(j)$, for $j \in S$. We have*

$$\sigma_{S,j}^2 = \sigma_j^4 \left( \sigma_j^2 - B_{U_j,j}^\top [\Theta_{\mathsf{an}(j)}]_{U_j,U_j}^{-1} B_{U_j,j} \right)^{-1}, \tag{10}$$

$$[B_S]_{k,j} = \frac{\sigma_{S,j}^2}{\sigma_j^2} \left( B_{k,j} - B_{U_j,j}^\top [\Theta_{\mathsf{an}(j)}]_{U_j,U_j}^{-1} [\Theta_{\mathsf{an}(j)}]_{U_j,k} \right). \tag{11}$$

**Proposition 1** *Denote the ancestors of $j \notin \mathcal{I}$ in $\mathcal{I}$ by $\mathsf{an}_{\mathcal{I}}(j)$. If a set $S$ contains $\mathsf{an}_{\mathcal{I}}(j)$ and their parents $\mathsf{pa}(\mathsf{an}_{\mathcal{I}}(j))$, then $\sigma_{S,j}^{(1)} = \sigma_{S,j}^{(2)}$. Furthermore, for $i \in \mathcal{I}$ and any set $S$ we have $[\Delta_{\Theta_S}]_{i,i} \neq 0$. Additionally, if $[B_S]_{j,i} \neq 0$ in either model, then we have $[\Delta_{\Theta_S}]_{j,i} \neq 0$.*

**Remark 2** *We repeatedly use the restricted SEM characterization in Lemma 1 with various strategic choices of subsets $S$ in Algorithm 1 to eliminate the non-intervened nodes from $S_\Delta$ using the criterion of Proposition 1. In Step 1, to identify $J_0$ in (7), we set $S = \{j\}$ for each $j \in S_\Delta$. In Step 2, to identify $J_0^k$ in (8) for each $k \in S_\Delta \setminus J_0$, we set $S = \{j, k\}$ for each $j \in J_0$. In Step 3, to process the nodes in $\mathcal{A}_\ell$, we use subsets of the form $\mathcal{M}_\ell \cup A$ for subsets $A$ in $\mathcal{A}_\ell$.*

**Theorem 1 (Consistency)** *If Assumption 1 holds and covariance estimates are perfect, then Algorithm 1 consistently estimates soft interventions with $\mathsf{P} = 1$.*

### 4.2 $\mathcal{I}$-Markov equivalence

Interventions in a DAG change the conditional distributions of the intervened variables, and hence, they reveal orientations of some edges that were previously undirected in observational CPDAG, resulting in the interventional CPDAG ($\mathcal{I}$-CPDAG). The DAGs that have the same $\mathcal{I}$-CPDAG under soft intervention $\mathcal{I}$ form the $\mathcal{I}$-Markov equivalence class ($\mathcal{I}$-MEC). This is shown and discussed next.

For a DAG $\mathcal{G}$ and an intervention set $\mathcal{I}$, an additional $\mathcal{I}$-vertex $\zeta$ and corresponding $\mathcal{I}$-edges $\{\zeta \to i\}_{i \in \mathcal{I}}$ are added to form the interventional DAG ($\mathcal{I}$-DAG). Note that vertex $\zeta$ creates a new v-structure $\zeta - i - j$ for any non-intervened $j \in \mathsf{pa}(i)$. However, if $j$ is also in $\mathcal{I}$, then $\mathcal{I}$-DAG also contains the $\zeta \to j$ edge, and there is no new v-structure that can orient the edge $i - j$.

We call the edges in $\mathcal{G}$ that are not directed in the original CPDAG but are directed in $\mathcal{I}$-CPDAG as $\mathcal{I}$-directed edges. In the *parent finder* step of Algorithm 1, we find the edge set $\{j \to i\}_{j \notin \mathcal{I}, i \in \mathcal{I}}$, and subsequently, obtain the $\mathcal{I}$-MEC. Therefore, we can use Algorithm 1 in conjunction with an observational algorithm to perform causal structure learning, and establish the following theorem.

**Theorem 2 ($\mathcal{I}$-MEC)** *If Assumption 1 holds and covariance estimates are perfect, then Algorithm 1 consistently recovers non-intervened parents of an $i \in \mathcal{I}$ with $\mathsf{Q} = 1$. This result modifies the original MEC, which can be obtained via any observational structure learning algorithm, into the $\mathcal{I}$-MEC.*

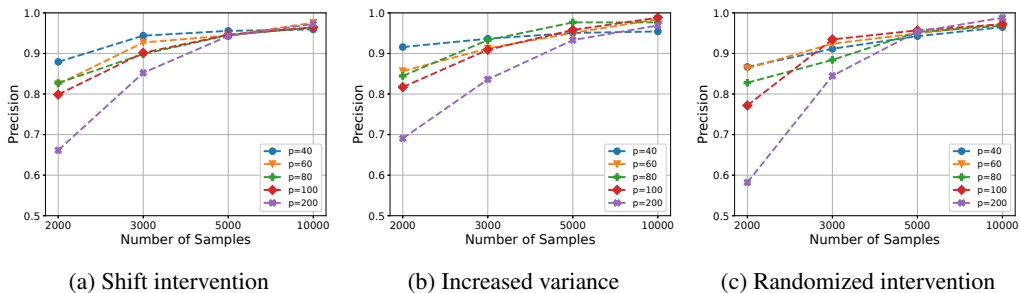

| (a) Shift intervention | (b) Increased variance | (c) Randomized intervention |

Figure 1: Average precision of estimating intervention targets. Algorithm 1 reaches high precision with increasing number of samples even in large models for all settings.

### 4.3 Sample complexity

In this subsection, we provide the finite-sample counterparts of Theorem 1 and Theorem 2. Our choice of the PDE algorithm, the ADMM-based method of [12], enjoys finite-sample results when noise $\epsilon$ has a Gaussian distribution. The following theorem establishes the sample complexity of Algorithm 1 for estimating $\mathcal{I}$ and the non-intervened parents of the nodes in $\mathcal{I}$.

**Theorem 3 (Sample complexity)** *Let $d$ denote the maximum degree of an intervened node and set $\Gamma \triangleq \Sigma^{(2)} \otimes \Sigma^{(1)}$ and $\alpha \triangleq 1 - \max_{e \notin \mathrm{supp}} |\Gamma_{e,\mathrm{supp}} \Gamma_{\mathrm{supp},\mathrm{supp}}^{-1}|_1$. Accordingly, define $M \triangleq \max\{\|\Sigma^{(1)}\|_\infty, \|\Sigma^{(2)}\|_\infty\}$, $M_\Sigma \triangleq \max\{\|\Sigma^{(1)}\|_{1,\infty}, \|\Sigma^{(2)}\|_{1,\infty}\}$, $M_{\Gamma,\Gamma^T} \triangleq \max\{\|\Gamma_{S,S}\|_{1,\infty}, \|\Gamma_{S,S}^T\|_{1,\infty}\}$, where $S$ is the support of $(\Sigma^{(2)})^{-1} - (\Sigma^{(1)})^{-1}$. When $\alpha > 0$ and $M_\Sigma M_{\Gamma,\Gamma^T} < +\infty$, with $n = O\left(\frac{d^4}{\varepsilon^2} \frac{\log p}{\delta}\right)$ samples, Algorithm 1*

1. *identifies $\mathcal{I}$ with a probability at least $\geq 1 - \delta$;*

2. *identifies the non-intervened parents $\{j \to i\}_{j \notin \mathcal{I}, i \in \mathcal{I}}$ with a probability at least $\geq 1 - \delta$.*

Note that we have assumed that the product $M_\Sigma M_{\Gamma,\Gamma^T}$ is bounded. This is necessary to avoid a linear scaling of the sample complexity in $p$. More discussion on the necessity and implications of this assumption is provided in the proof of Theorem 3 in Appendix A.

## 5  Empirical results

### 5.1  Synthetic data - intervention recovery

We start by testing our algorithm for estimating intervention targets, i.e., the set $\mathcal{I}$. We generate 100 realizations of Erdős-Rényi [21] DAGs with expected neighborhood size $c = 1.5$, and $|\mathcal{I}| = 5$. We sample the entries of $B$, i.e., the edge weights, independently at random according to the uniform distribution on $[-1, -0.25] \cup [0.25, 1]$. The additive Gaussian noise terms have distribution $\mathcal{N}(0, I_p)$. We select the intervention set $\mathcal{I}$ by randomly selecting 5 nodes from $[p]$. We consider three different models to intervene on the nodes in $\mathcal{I}$: (i) *shift intervention* model in which mean of the noise $\epsilon_i$ is shifted from 0 to 1, (ii) *variance increase* model in which the variance of the noise $\epsilon_i$ is increased from 1 to 2, and (iii) *randomized intervention* model, in which $(B^{(2)})_{\mathrm{pa}(i),i} = 0$ and the noise variance varies from 1 to 1.5. All the simulations are run on a MacBook Pro with 2.7 GHz Dual-Core i5 core and 8 GB RAM.

We first run our algorithm by varying the graph size $p$ and the number of samples. Figure 1 illustrates that our algorithm is able to recover the intervention targets with high precision under all three intervention models. Having high precision is especially important in high dimensions, since a large false positive rate severely affects any downstream task such as structure learning. Recall rates are close to 1 and they are omitted from the graph.

Next, we compare our results with that of the UT-IGSP algorithm [19] for the shift intervention model. We note that UT-IGSP performs a greedy search to identify the sparsest permutation through CI tests, and it returns intervention targets as a by-product along with the learned causal structure. The computation time of UT-IGSP, hence, grows quickly with the size of the graph, reaching an

average of 61.2 seconds for $p = 100$. Therefore, the complexity of structure learning and intervention target discovery in the high-dimensional regime is prohibitive. In contrast, our algorithm has comparable performance to UT-IGSP when $p = 100$, while requiring less than a second of runtime. Our algorithm's runtime scales gracefully when the dimension is in the hundreds.

Table 1: $\mathcal{I}$ estimation in the shift intervention model - 50 repetitions with 5000 samples - density 1.5

| | UT-IGSP ( [19]) | | | Algorithm 1 | | |
|---|---|---|---|---|---|---|
| p | Precision | Recall | Time(s) | Precision | Recall | Time(s) |
| 40 | 0.99 (0.04) | 0.99 (0.04) | 0.8 | 0.96 (0.09) | 0.94 (0.09) | 0.1 |
| 60 | 0.95 (0.07) | 0.99 (0.05) | 5.2 | 0.97 (0.07) | 0.95 (0.10) | 0.2 |
| 80 | 0.96 (0.08) | 0.99 (0.04) | 17.8 | 0.96 (0.08) | 0.96 (0.10) | 0.3 |
| 100 | 0.93 (0.11) | 1 (0) | 61.2 | 0.94 (0.09) | 0.98 (0.07) | 0.3 |

## 5.2 Synthetic data - causal structure learning

In Section 4.2, we have shown that Algorithm 1 recovers the intervention targets. It can be further used to refine the observational MEC to an $\mathcal{I}$-MEC. Accordingly, we take the correct CPDAG of $\mathcal{G}^{(1)}$, and then apply our algorithm's findings to obtain $\mathcal{I}$-CPDAG. We report the accuracy of additional edge orientations and in particular recovery of parents (if possible) of intervention targets in Appendix B.2.

## 5.3 Application to real data

We apply our algorithm to two real datasets with observational and interventional data to learn their causal structures. When there exist multiple interventional environments, we apply our algorithm to them individually to estimate the intervened nodes and their parents. Subsequently, we combine the results from all environments in order to form the final estimated structure. There is a large number of interventional settings in which finding the targets and their non-intervened parents by our algorithm yields a good estimate of the associated DAG, which we use for our evaluation.

We compare our results with those of the algorithms UT-IGSP, and UT-IGSP* in [19][1], where the former works with partially known intervention targets and the latter does not require any target input. We use both parametric (Gaussian) and non-parametric (Hilbert-Schmidt independence criterion) CI tests for UT-IGSP methods. We note that the non-parametric tests have significant runtimes. We note that our algorithm uses PDE at several stages, which calls for different $\lambda$ regularization parameters. Namely, let us denote the parameters used for Step 1 and Step 3 by $\lambda_1$, Step 2 by $\lambda_2$ and the *parent finder* of Step 3 by $\lambda_3$. Similarly, UT-IGSP needs a cut-off value $\alpha$ for CI tests. We run the algorithms with different values of these parameters to obtain the receiver operating characteristic (ROC) curves.

**Protein signaling data.** We first consider the dataset in [22] for discovering the protein signaling network of 11 nodes. It consists of measurements of proteins and phospolipids under different interventional environments. In each environment, signaling nodes are inhibited or activated. Hence, these sites form intervention targets. The conventionally accepted ground truth has been updated over the years, and we compare with the recent version in [23], which consists of 16 edges. We follow the process of [19] and work with 1755 observational and 4091 interventional samples aggregated from 5 different interventional environments. In Fig. 2a, we report the results of running Algorithm 1 and UT-IGSP with various parameters.

**Perturb-seq gene expression data.** We analyze the performance of our algorithm on the perturb-seq dataset by in [24]. The dataset consists of observational data and the interventional data from bone marrow-derived dendritic cells (BMDCs). A single gene has been targeted for deletion in each interventional environment. Similarly to [24], we have focused on 24 genes that are known to have regulatory effects, and we have followed [16] to select interventional data from 8 gene deletions along with observational samples. We take the Fig. 4D in [24] as the ground truth, which has 34

---

[1]The code and preprocessed real data are taken from https://github.com/csquires/utigsp for fair comparison, and CausalDAG package which is distributed under 3-Clause BSD licence is used.

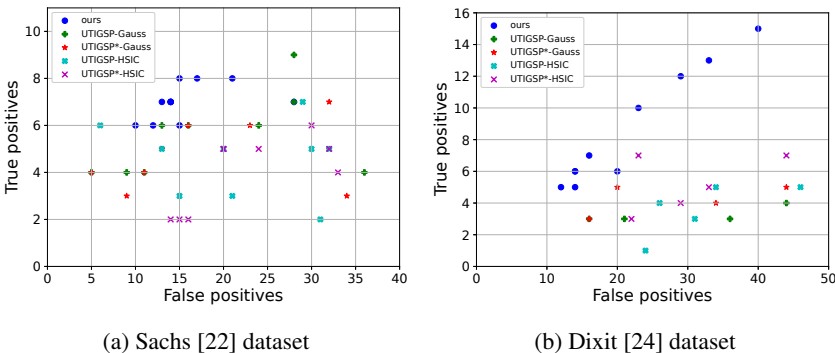

(a) Sachs [22] dataset

(b) Dixit [24] dataset

Figure 2: ROC curves for directed edge recovery. Algorithm 1: (a) is more robust in smaller graphs; (b) handles larger graphs and multiple interventional settings successfully, and conforms to real datasets better than CI testers.

edges among 24 nodes. We use 23 interventional settings for the targeted 8 gene deletions. In Fig. 2b, we plot the results of running Algorithm 1 and UT-IGSP with various parameters.

In both real datasets, our algorithm achieves higher accuracy in recovering directed edges. The comparison with UT-IGSP is more striking in Fig. 2b, and shows that our ability to work with many interventional environments on a relatively larger graph. Furthermore, Fig. 2a shows that Algorithm 1 handles smaller graphs more robustly. These results illustrate that even though our algorithm is designed for linear models, on real datasets, it performs better than the current state-of-the-art methods that rely on CI tests.

## 6 Conclusion

In this paper, we have considered the problem of estimating intervention targets in linear structural equation models (SEMs) under soft interventions. We have proposed an algorithm that consistently identifies intervened nodes that can scale to larger graphs and have sample complexity guarantees in Gaussian linear SEMs. The algorithm can be used also to infer interventional Markov equivalence class (MEC) from the observational MEC. We have demonstrated comparable or better performance compared to the existing methods in a number of settings.

The limitation of our method is that it only applies to linear SEMs. The dataset in an application should be evaluated carefully to confirm whether the assumptions are satisfied. This avoids any adverse effects arising from wrong interpretations of cause-effect relationships. Extending the similar ideas for scalable and efficient intervention target estimation to the non-linear DAGs is an open question that we aim to address in future work. Finally, the condition number of the optimization problem is assumed to be bounded in the sample complexity results. We note that our algorithm is independent of the specific precision difference estimation (PDE) algorithms and can be used in a modular way. In this regard, it can benefit from any potential relaxation on this limitation of PDE algorithms.

## Acknowledgments and Disclosure of Funding

This work was supported by the Rensselaer-IBM AI Research Collaboration (http://airc.rpi.edu), part of the IBM AI Horizons Network (http://ibm.biz/AIHorizons).

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
