# OpenReview forum: "Scalable Intervention Target Estimation in Linear Models"
_NeurIPS.cc/2021/Conference — NeurIPS 2021 Poster_

### Official Review · Reviewer_ovCV · 2021-07-16

**Rating:** 6
**Confidence:** 3

**Summary:**

This paper considered estimating the intervention target nodes in the linear structural equation model with Gaussian noises. The proposed method was built upon the difference in precision matrices across observational and interventional datasets, under the soft intervention setting and sparsity assumption. The authors provided a comprehensive discussion on their algorithm including computational complexity, theoretical analysis, and sample complexity. My main concerns lie in their motivation (about intervention recovery), novelty comparing with existing literature, and utility of the soft intervention setting.

**Limitations And Societal Impact:**

The authors claimed their method is limited to linear SEMs. However, the sparsity and normality assumptions are also limitations. The authors could consider improving their motivations and enhancing their contributions, as detailed in the main review.

**Main Review:**

This paper focused on identifying the intervention target nodes in the linear structural equation model, by defining a soft intervention setting and leveraging the ADMM-based method proposed in B. Jiang et al (2018). The sparsity and normality assumptions are required to guarantee computational efficiency and theoretical consistency. I have several concerns regarding their motivation (about intervention recovery), novelty comparing with existing literature, and utility of the soft intervention setting.

1. *Motivation on Intervention Recovery*: In this paper, the authors care about intervention recovery and causal discovery based on both interventional and observational datasets. I agree the latter one is one of the most important topics in causal structural learning, while why do we care about the unknown interventions? Just one sentence regarding gene-editing example was included in the current paper. The authors should discuss more as it may be the main contribution of this work.

2. *Novelty Comparing with Existing Literature*: The main idea of this paper is to estimate the intervention nodes from the difference in precision matrices across observational and interventional datasets. There were actually many similar/related works proposed while haven't been discussed in this paper. For instance, Ghassami, AmirEmad, et al (2020) considered causal equivalence of two DAGs *also based on the precision matrices*.  Jaber, Amin, et al (2020) formulated graphical characterization to test whether two pairs of causal graphs and their corresponding interventional target sets. The authors should discuss the contributions beyond these works.

-- Ghassami, AmirEmad, et al. "Characterizing distribution equivalence and structure learning for cyclic and acyclic directed graphs." International Conference on Machine Learning. PMLR, 2020.

-- Jaber, Amin, et al. "Causal discovery from soft interventions with unknown targets: Characterization and learning." Advances in neural information processing systems 33 (2020).

3. *Utility of Soft Intervention Setting*: This paper considered a soft intervention setting that does not remove causal effects on intervention target nodes and claimed that this setting is more practical. It seems such a setting is more friendly to develop theory and related methods, but less related to utility. It will be better to elaborate more on this with some concrete examples.





**Time Spent Reviewing:**

5 hours

---

> ### Author Response · Authors · 2021-08-10
> **We address the valid concerns on the importance of unknown interventions, elaborate on the distinctions of our work from existing literature, and clarify the utility of the soft intervention setting.**
>
> We are grateful for the thorough and thoughtful evaluations and comments.
>
> **Motivation on intervention recovery**:
> We thank the reviewer for pointing out the need for elaboration on the motivation of recovering unknown interventions. We provide a few more concrete examples below and will discuss them in the final version. Besides enabling a broader range of algorithms, such as IGSP [11] or GIES [10], to learn from interventional data, recovering intervention sites alone is an important problem. For instance, in EEG analysis, it is of interest to detect neurons or different brain regions that interact differently when the subject is performing various tasks (Sanei and Chambers (2013), and [4]). These neurons or brain regions correspond to intervened nodes in a causal graph representation. Another application area is large-scale systems. There are recent papers in which researchers model IoT and cloud applications using causal graphs (see survey paper of Bogatinovski et al. (2021)). Even though it is not fully applied in this context yet, faulty nodes in the system can be considered as intervened nodes and localized through our methods. In this area, fault localization is a very important problem and some relevant work can be found in Section 3.2 of Bogatinovski et al. (2021).
>
> -- S. Sanei and J. A. Chambers. EEG Signal Processing. John Wiley \& Sons, 2013.
>
> -- Bogatinovski, Jasmin, et al. "Artificial Intelligence for IT Operations (AIOPS) Workshop White Paper." arXiv:2101.06054 (2021).
>
> **Novelty comparing with existing literature**: We provide the distinctions from the mentioned papers. We will add the necessary elaboration in the final version.
> - Our problem settings, objectives, and solutions are quite distinct from the work of Ghassami et al. (2020), which uses precision matrices to formulate the equivalence classes. We note that its contribution is characterizing equivalence classes, and its learning algorithm is strictly for observational data. This is in sharp contrast to our focus on the interventional data. Indeed, we can always use the algorithm of Ghassami et al. (2020), or any other observational algorithm, to estimate the MEC, and use our algorithm on top of it to estimate $\mathcal{I}$-MEC and further improve scalability.
> - The main contribution of Jaber et al. (2020) is introducing a graphical characterization for the more complex case of latent confounders. However, their algorithm is based on conditional independence tests and a modification of FCI. Hence, it is not scalable. Figure 6 in the supplementary file of Jaber et al. (2020) shows that even with 6 nodes, their algorithm needs in the scale of 50 thousand samples before it reaches to moderate precision. It would be interesting to scale up their approach in the causally insufficient setting. Currently, we are addressing a different problem and provide scalable algorithms with sample complexity results for causally sufficient linear systems, that are lacking in the work Jaber et al. (2020) for their more complex problem.
>
> -- Ghassami, AmirEmad, et al. "Characterizing distribution equivalence and structure learning for cyclic and acyclicdirected graphs." International Conference on Machine Learning. PMLR, 2020.
>
> -- Jaber, Amin, et al. "Causal discovery from soft interventions with unknown targets: Characterization and learning." Advances in neural information processing systems 33 (2020).
>
> **Utility of soft intervention setting**: We clarify that our soft intervention setting does not require preserving causal effects on target nodes. In other words, we are not making any assumption on whether the causal effects on target nodes are changed or not - they can change freely. We believe this is a more realistic assumption. It has been known that removing the causal effects completely as in perfect interventions, or assigning certain values to the target nodes for hard interventions, can be difficult in practice. For instance, the effects of the infused chemicals to the cell are not set to specific values nor they are precisely known (see [12] and Jaber et al. (2020)). In this regard, we believe the considered soft intervention setting is a realistic and useful one.
>
> **Limitations**: The assumptions made in our work are the standard ones also made by the related work in [4] and [9]. We also provide sample complexity results, which are lacking in [4], and scalability results with soft interventions which are lacking in [9] lacks. In that sense, linear models with sparsity and norm assumptions are used to obtain the novel sample complexity results presented. In the final version, we will add a comment on the assumptions to the section on our limitations.

---

> > ### Comment · Reviewer_ovCV · 2021-08-25
> > **Post-Rebuttal**
> >
> > I appreciate the authors' time and effort in addressing my comments. I think the authors have done a good job of addressing my concerns. I have updated my score accordingly.

---

### Official Review · Reviewer_LsC2 · 2021-07-16

**Rating:** 6
**Confidence:** 2

**Summary:**

Given two datasets generated from two DAGs, where the second DAG is assumed to be created from the first DAG by varying the noise level in a subset of nodes (called intervened nodes) in the linear structure equation model, this paper proposed an efficient algorithm to identify the subset of intervened nodes and their parents.

The proposed algorithm consists of three steps. First, apply precision difference estimation (PDE) to identify the set of non-intervened nodes. This is feasible based on the established fact that the diagonal entry of the precision matrix difference is non-zero if and only if the corresponding node or its child receives intervention. Second, from the set of non-intervened nodes identified from step 1, form equivalent classes. Third, those equivalent classes are further processed in step 3.

The computational complexity of PDE is cubic w.r.t the number of nodes. In step 1, PDE nees to be run multiple times. Furthermore, in step 3, the complexity could be exponential. The algorithm is provided to be consistent in Theorem 1. The sample complexity bound is also established in Theorem 3.

In the experimental results, the proposed algorithm is shown to be able to recover the ground truth set of intervention targets with high precision given large number of samples. It's also much faster than its competing alternative of UT-IGSP algorithm. the proposed algorithm is also further evaluated on real-world datasets, including protein signaling data and perturb-seq gene expression data.

**Ethics Review Area:**

["I don’t know"]

**Limitations And Societal Impact:**

I didn't see any negative societal impact of this work.

**Main Review:**

To the best of my knowledge, the major contribution of this work, i.e. scaling up the identification of intervened nodes given a pair of datasets, seems novel. The experiments show it runs much faster than the baseline approach UT-IGSP.

Will it be difficult to implement the soft intervention in (4)? Does this require the identification of the first DAG structure to determine its noise level?

The exponential complexity of step 3 is still worrisome. Can the authors present some edge cases where the algorithm might get stuck in this step when the size of A_l becomes too large?

In section 5.1, the precision seems much lower when the number of samples is 2000. In real-world datasets, does this mean the number of intervened samples have to be at the scale of thousands?

In the experimental section, besides UT-IGSP algorithm, is there any there baseline approach the proposed algorithm can compare against? To establish the efficiency of the proposed algorithm, it might be more convincing to compare it against at least a few benchmark algorithms.

Is the ADMM solver for the PDE in Equation (6) the most efficient algorithm? Since PDE will be used multiple times in step 1 and 2, it might be useful to explore a more efficient solver of the PDE. Cubic complexity w.r.t. the number of nodes means the algorithm cannot scale to thousands of nodes, correct?


**Time Spent Reviewing:**

2

---

> ### Author Response · Authors · 2021-08-10
> **We explain the availability of soft interventions in real settings, elaborate on the scalability of our algorithm and its ability to work with limited data in real data, and point out comparisons we have made in Appendix.**
>
> We thank the reviewer for asking thoughtful questions that motivate us to explain our algorithm in more details. We hope that our explanations below sufficiently address these questions.
>
> **Soft interventions** :
> We would like to clarify two issues:\
>     1 - In this paper, we are not implementing interventions. There is a separate line of work, e.g., [18] and Singer et al. (2016), which focus on performing interventions in various ways. Instead, we are collecting the interventional data from the nature.  \
>     2 - Soft interventions are easier to observe in real world. We provide some examples of the soft interventions that occur in various natural and technological domains.
> - The elements of an advertising system (e.g., query, prices, and clicks) can be formulated using a causal graph (Fig. 12 of Bottou et al. (2013)). Interventions to this system are just modifications to the intermediate nodes, which do not remove causal effects. Hence, the interventions in such systems (e.g., approach of Bottou et al. (2013) to answer counterfactual questions in real search engines) are soft interventions.
> - Another example is the protein signaling network in [18], which is one of the real datasets that we have used in Section 5.3. The proteins are intervened by injecting small-molecule inhibitors to the cell. Such interventions are not hard, i.e., they do not remove the causal effects in the protein signaling network. Rather, they are soft interventions that only modify these effects.
>
> -- Singer, Meromit, et al. "A distinct gene module for dysfunction uncoupled from activation in tumor-infiltrating T cells." Cell 166.6 (2016): 1500-1511. \
> -- Bottou, Léon, et al. "Counterfactual Reasoning and Learning Systems: The Example of Computational Advertising." Journal of Machine Learning Research 14.11 (2013).
>
>  **Exponential complexity**:
> We elaborate on the computational complexity of Step 3 in various aspects.
> - The size of the largest equivalence class is significantly smaller than the size of the set of the changed nodes, $S_\Delta$, in most cases, and the algorithm does not get stuck. In Appendix B.4, we discuss this relation in more details, and provide simulation results in Fig. 4 to support our claim.
> - One possible scenario for the size of $\mathcal{A}_\ell$ to become prohibitive is the parents of intervention targets are also intervened. In this case, $J_0$ will be the empty set, and all nodes in $S_\Delta$ will belong to the same group. However, this requires the interventions to concentrate in one neighborhood such that parents of the intervened nodes will also be intervened. In reality, such scenarios happen rarely, and interventions are generally distributed.
> - Finally, we also emphasize the exponential complexity of the existing studies. For instance, on the related subject of difference estimation, the algorithm in [4] has exponential complexity with $|S_\Delta|$. The algorithm in [9] is claimed to be more efficient, but does not provide a formulated advantage over $|S_\Delta|$.
>
> **Number of samples**:  We thank the reviewer for pointing a valid concern from Fig. 1. In Section 5.1 simulations, we have mainly focused on the scalability of our algorithm, and have used 5 intervention targets in each case, without aiming for greater precision with fewer number of samples in simpler settings. In Table 6 of Appendix B.2, we have provided a comparison in terms of the increasing number of intervention targets. This sheds more light on the the difficulty of the considered setting in Section 5.1. More importantly, we point out our real data experiments in Section&nbsp;5.3. For instance, the perturb-seq gene expression dataset has 23 different interventional settings, where the number of samples vary from 126 to 1938, and only 4 of these settings have more than 1000 samples. Our algorithm runs for each of these settings individually without needing thousands of samples.
>
> **Baseline algorithms**: We note that most of the literature on learning from interventional data requires known intervention targets, e.g. IGSP [11] and GIES [10]. Therefore, these algorithms solve a different problem and are not applicable to our problem. There are two other alternatives to work with unknown interventions beyond UT-IGSP. First is the algorithm for difference learning presented in [9]. We explain how we can modify that algorithm to our problem in Appendix B.1 and provide the comparison in Table 5. The second one is the DCDI algorithm of [14]. We note that runtime of DCDI exceeds 7 hours with as few nodes as 50 (see Fig. 10 in the supplementary file of [14]) while ours takes less than a second. Hence, we did not consider comparing our algorithm against DCDI.
>
> **Complexity of PDE function**: We have explored several options for the PDE function, and have concluded that the ADMM-based algorithm of [7] is the best option. We note that a related study in [9] uses another PDE function, which has complexity $O(p^4)$, and reducing it to $O(p^3)$ is a significant gain, which allows us to process hundreds of nodes. Furthermore, we need PDE on $p$ nodes only once, during the $S_\Delta$ estimation in Step 1. The rest of the PDE instances require much smaller number of nodes as stated in Remark 3 in Appendix A. We also remark that the main computational improvement of our algorithm is reducing the number of PDEs. The existing interventional learning algorithms such as UT-IGSP cannot even scale to hundreds of nodes efficiently (see Table 1). We provide the average runtime for larger Erdos-Renyi random graphs with density $c=2.5$ and $|\mathcal{I}|=5$ over 10 graph instances below.
>
>       number of nodes p   -   average runtime (seconds)
>                    100    -   0.5
>                    250    -   3.1
>                    500    -   19.8
>                   1000    -   132.7

---

> > ### Comment · Reviewer_LsC2 · 2021-08-17
> > **Response to rebuttal**
> >
> > I'd like to thank the author for addressing my comments in detail. I'm convinced by these explanations and will increase my rating to recommend the acceptance of this work.

---

### Official Review · Reviewer_wJ1B · 2021-07-19

**Rating:** 7
**Confidence:** 5

**Summary:**

The paper consider the problem of learning the interventional Markov equivalence class of causal directed acyclic graph from observational and interventional data where the intervention targets are unknown and there are no unobserved variables. This problem and its variation havve been previously studied by Squires, Wang, and Uhler (2020) among others and is an important problem in the causal inference literature. The main contribution of the paper is an algorithm that is significantly faster than the permutation search algorithm (UT-IGSP) of Squires, Wang, and Uhler 2020 while also having finite sample guarantees which is lacking for UT-IGSP.  However, the paper makes slightly stronger assumptions than the UT-IGSP algorithm in specific form of interventions allowed: that the interventions must change the noise variances, and only works for linear SEMs. The authors also demonstrate the superiority of their method on rigorous synthetic and real world experiments.

**Limitations And Societal Impact:**

Yes. The authors mention that they assume linear models and that this must be checked in real world data to ensure causal conclusions are not misinterpreted.

**Main Review:**

# Strengths
- The paper is well written and the problem is well motivated.
- Technically, the authors largely build on the work of Ghoshal and Honorio (2019) for directly learning the difference between structural equation models (SEMs) to learn the set of intervened nodes and their parents. While Ghoshal and Honorio (2019)'s algorithm was proposed for the problem of directly estimating the difference DAG of two SEMs under the assumption that noise variances are invariant between the two linear SEMs, one of the key contributions of the paper is to recognize that the main ideas of the algorithm lends itself well to the problem of estimating the target of interventions from observational and interventional data without significant assumptions on the interventional setting.
- While the main steps of the algorithm are essentially the same as Ghoshal and Honorio (2019)'s algorithm: estimate difference of precision matrices, remove vertices for which the difference of precision matrices has all zeros, then construct an ordering over subsets of remaining vertices, and finally remove edges (in this case vertices) by estimating difference between precision matrices over subsets of vertices; the authors also leverage additional non-trivial insights from the problem like (i) some of the non-intervened targets can be estimated from the covariance matrices, (ii) the ordering can be estimated by only estimating difference of precision matrices over only sets of two nodes, and (iii) finally constructing orderings over equivalence classes, to propose an algorithm whose computational complexity is exponential only over the size of the largest equivalence class. The size of the largest equivalence class can be significantly smaller than the set of changed nodes (in the difference of precision matrices).
- Another improvement over Ghoshal and Honorio (2019) is the choice of algorithm used to estimate the difference between precision matrices which has a complexity of $O(p^3)$ instead of the $O(p^4)$ used by the former.
- The authors obtain finite sample guarantees for their method and obtain conditions for high-dimensional recovery.
- Synthetic experiments demonstrate that the proposed algorithm is significantly faster than UT-IGSP while being comparable in accuracy. Real world experiments are also insightful and demonstrate the applicability of the method even though the method was designed for linear SEMs.

# Weaknesses
1. Example 1 is incorrect. Node 2 should not be part of $J_0$ since node 2 is not part of $S_{\Delta}$.
2. The high-dimensional results are obtained under the assumption that the $\ell_{1}/\ell_{\infty}$ norm of  $\Sigma^{(1)}$  is constant which is unreasonable since the covariance matrices are generally dense (especially under the assumption that the individual SEMs are dense) and the $\ell_{1}/\ell_{\infty}$ norm is O(p). In comparison, Ghoshal and Honorio (2019)'s method depends on  $||\Sigma^{(1)}||_{\infty}$ (element-wise infinity norm).

# Other comments
- $\ell_{1}/\ell_{\infty}$ group norms are not defined and it is not clear that $||.||_{\infty}$ is the element-wise norm of a matrix.
- Line 163: the notation $an_{I}(j)$ has not been defined at this point to mean the ancestors of j in the set $I$.

**Time Spent Reviewing:**

3

---

> ### Author Response · Authors · 2021-08-10
> **We agree with the reviewer's comment on the growth of the $\ell_1 / \ell_\infty$ norm of the covariance matrix. We explain the practical implication of the assumption, and address the minor clarification issues.**
>
> We agree with the reviewer that the $\ell_1 / \ell_\infty$ norm of the covariance matrix can grow with $p$. After Theorem 3 we have explained that we need them to be bounded for the result to not grow linearly with $p$. As the reviewer has pointed out, our algorithm uses a PDE function, that has complexity $O(p^3)$, whereas the PDE function used in [9] has complexity $O(p^4)$. Even though the theoretical guarantees hold with slightly more restrained conditions, we find this improvement necessary for scalability. Indeed, it is important to improve the theoretical and computational results of PDE functions in future research.
>
> We thank the reviewer for pointing other minor issues. In Example 1, the link $2 \rightarrow 5$ is missing in the figure. It causes $2$ to be in $S_\Delta$, and results in the groups decomposition in the text. We will fix the figure and define the norms and intervened ancestors clearly in the final version.

---

> ### Comment · Area_Chair_d6DG · 2021-09-10
> **A possible problem with the sample complexity**
>
> I am looking at the comment from reviewer wJ1B about $M\_\\Sigma = \\max( \\| \\Sigma^{(1)} \\|\_{1,\\infty} , \\| \\Sigma^{(2)} \\|\_{1,\\infty} )$ which is $O(p)$ for dense matrices with $O(1)$ entries.
> By looking at the proof of Theorem 3, it uses Corollary 1 of [7], which is based on Theorem 1 of [7] which shows that the sample complexity is $O(M\_\\Sigma^4 M\_{\\Gamma,\\Gamma^T}^4 d^4 \log p)$, not just $O(d^4 \log p)$.
>
> The term $M\_{\\Gamma,\\Gamma^T}$ in [7] depends on the inverse of a submatrix (and submatrix of the inverse) of a Kronecker product. More formally $M\_{\\Gamma,\\Gamma^T} = \\max( \\| \\Gamma\_{S,S}^{-1} \\|\_{1,\\infty} , \\| (\\Gamma\_{S,S}^T)^{-1} \\|\_{1,\\infty} )$, where $S$ is the support of the precision matrix $(\\Sigma^{(2)})^{-1} - (\\Sigma^{(1)})^{-1}$ and $\\Gamma = \\Sigma^{(2)} \\otimes \\Sigma^{(1)}$.
>
> The fact that these dependences have not been discussed in depth (even in the authors' rebuttal) makes me a bit uncomfortable. In fact, the term $M\_{\\Gamma,\\Gamma^T}$ was never mentioned. I am struggling to see which scenario is reasonable for the above. Let me explain:
>
> - For two given dense matrices $\\Sigma^{(1)}$ and $\\Sigma^{(2)}$ with $O(1)$ entries, if I try to make $M\_\\Sigma$ to be $O(1)$ let say by dividing the matrices by $p$, then $M\_{\\Gamma,\\Gamma^T}$ will become $O(p^2)$.
>
> - For two given dense matrices $\\Sigma^{(1)}$ and $\\Sigma^{(2)}$ such that $M\_\\Sigma$ is $O(1)$, then the entries of $\\Sigma^{(1)}$ and $\\Sigma^{(2)}$ go to zero as $p$ goes to infinity.
>
> - For two given sparse matrices $\\Sigma^{(1)}$ and $\\Sigma^{(2)}$, the problem already has a lot of independence since Gaussianity is assumed (in the covariances, not in the difference of the precision matrices).
>
> [7] https://jmlr.org/papers/volume19/17-285/17-285.pdf

---

> > ### Author Response · Authors · 2021-09-11
> > **author response**
> >
> > Thank you for the thoughtful comments. We would like to highlight the following issues.
> >
> > 1. **General sample complexity**: Indeed, as the AC correctly has pointed out, in the most general form, the sample complexity does depend on the terms $M_\Sigma$ and $M_{\Gamma,\Gamma^T}$.
> >
> > 2. **Sample complexity under our assumptions**: Please note that under the assumptions of the problem considered these two terms do not affect the sample complexity. Specifically, as stated in the theorem and the very next sentence ensuing the theorem, **we are assuming that $M_\Sigma$ and $M_\Gamma$ are bounded.**
> >
> > 3. **Interpreting our assumption**: We appreciate the AC’s effort into finding insights into the above assumption. To clarify, we note the following:
> >
> >     a.  **Our assumption can be relaxed**: We can further relax our assumption that $M_\Sigma$ and $M_{\Gamma,\Gamma^T}$ are bounded, and instead assume that **only their product is bounded**. This product is, generally, called the **condition number of the estimation problem.**
> >
> >     b.  This relaxed assumption suffices to obtain the sample complexity $O(d^4 \log p)$. We will make this relaxation to the Theorem 3 conditions.
> >
> >     c.  This relaxed assumption brings about a significant level of flexibility in choosing $\Sigma^{(1)}$ and $\Sigma^{(2)}$. For instance, we can have **non-sparse choices for these matrices and still satisfy the relaxed assumption.**
> >
> >     d.  **Widespread precedence**: We would like to note that our relaxed assumption is a standard one in a broad range of matrix inference (e.g., estimation) problems. Specifically, the sample complexity of a large number of matrix estimation problems depends on the power of the condition number, and inevitably, it is needed to be bounded for finite-sample results. We provide two examples:
> >     - **Graphical lasso**: A similar term (condition number to power 2) appears in the graphical lasso literature as well (Ravikumar et al. (2008)). Please refer to page 8 for the definition of the condition number, which depends on i) the $\ell_\infty$-norm of the inverse of Hessian of the optimization problem, and ii) the $\ell_\infty$-norm of the covariance. Note that these are precisely the same quantities that appear also in our sample complexity, which scales with the product of these two quantities.
> >
> >     - **Matrix completion**: For another related but far-off example, please refer to page 6 of Jain et al. (2012) for matrix completion results. There is a 4th power dependence on the condition number measured in the 2-norm (this is a simpler case than estimation precision differences).
> >
> > 4. **How sensitive is our result to the relaxed assumption?** In a broader perspective, we emphasize two points:
> >
> >     a.  Our main contributions are not dependent on specific PDE tools used, and the PDE algorithm can be replaced with other ones in a modular way.
> >
> >     b.  Our sample complexity inherits its dependence on the condition number from the PDE algorithm that it uses. As we stated in our response to the reviewer’s comments, our algorithm can benefit from any potential improvement on PDE algorithms, including the dependence in the condition number. With the constant progress on PDE algorithms, the new ones can replace our PDE algorithm in a modular way. Subsequently, our dependence on the condition number can improve with the improvement of the PDE algorithms (and their dependence on the condition number).
> >
> > Jain et al. (2012): https://arxiv.org/abs/1212.0467
> >
> > Ravikumar et al. (2008): https://arxiv.org/abs/0811.3628

---

> > > ### Comment · Area_Chair_d6DG · 2021-09-22
> > > **Re: relaxed condition**
> > >
> > > The authors agree with my assessment and promise a relaxed condition based on the product $M\_\\Sigma M\_{\\Gamma,\\Gamma^T}$. As the current Theorem 3 uses Corollary 1 and Theorem 1 of [7], it is unclear whether the modification is possible (for instance, there are some assumptions somewhat buried in Lemma 3 and Lemma 4 of [7]). In addition, it is unclear to me, which other parts of the paper might be impacted with a change of conditions.
> > >
> > > Perhaps another way to express my concern is through the experimental results, which while good with respect to runtime, are not that good with respect to precision and recall. Take for instance Table 1 in the main paper, where the proposed method either gets worse precision and recall (p=40) or equal/better precision but a worse recall (p=60, 80, 100). In addition, there is no result showing 100% F1 score in Tables 2 to 7. While it is OK to report precision, recall and F1 score, at some point (given enough samples) precision, recall and F1 score should be 100% to back up the theorem statement on exact recovery (given enough samples).
> > >
> > > [7] https://jmlr.org/papers/volume19/17-285/17-285.pdf

---

> > > > ### Author Response · Authors · 2021-09-22
> > > > **Validity of the theoretical and numerical results**
> > > >
> > > > 1) Our relaxed assumption has absolutely no impact on the analysis provided. The relaxation only affects the choices of covariance matrices $\Sigma^{(1)},\Sigma^{(2)}$. This relaxation was made to answer the AC’s earlier question about the choices of covariance matrices and the validity of the assumptions.
> > > >
> > > >     To clarify the validity of the results, proof of Theorem 1 of [7] does not assume any bound on $M_\Sigma$ or $M_{\Gamma,\Gamma^T}.$ Corollary 1 of [7] immediately follows from our relaxed condition on the product $M_\Sigma M_{\Gamma,\Gamma^T}$. Assumptions of Theorem 1 of [7] verified the assumptions of the sub results, e.g. Lemma 3 and 4 of [7], used inside its proof (see page 27 of [7]).
> > > >
> > > >      Furthermore, our algorithm for finding intervention targets and the interventional-MEC is not affected by the choice of PDE algorithm, let alone the conditions of the sample complexity results of the PDE algorithm.
> > > >
> > > > 2) We want to emphasize that even without the relaxation the results presented are correct. This relaxation was made only for clarifying some of the AC’s questions and for accommodating a broader range of choices for covariance matrices, and subsequently $M_\Sigma$ and $M_{\Gamma,\Gamma^T}$.
> > > >
> > > >
> > > > 3) We would like to remind that in statistical inference, a general premise is that the decisions are never perfect. Hence, expecting 100% accuracy for F1 is impossible. Furthermore, we simulate at least 50 independent graphs in each scenario. 100% F1 score would require each of the high probability PDE results to be correct, which is impossible for large number of trials. We report that our algorithm outperforms the state of the art for larger graphs (Tables 2-4), and meets the scalability promise of the paper.

---

### Decision · Program_Chairs · 2021-09-27

**Decision:**

Accept (Poster)

**Comment:**

The paper proposes an algorithm for estimating the unknown intervention targets in a causal linear structural equation model (SEM) with Gaussian noise. The paper considers observational and interventional data generated under soft interventions.

This paper was viewed as a borderline paper.

The authors learn the set of intervened nodes (and their non-intervened parents) by building on the work of directly learning the difference directed acyclic graph between SEMs by Ghoshal and Honorio (2019). The main steps of the proposed algorithm are essentially the same as Ghoshal and Honorio (2019)'s algorithm based on estimating the difference of undirected graphical models (inverse covariance matrices). In addition to that, the authors use non-trivial insights from the problem at hand.

If accepted, I recommend to include the following clarifications and modifications:
- I will give the benefit of the doubt regarding experiments. Please try a relatively larger number of samples, so that the F1 score is closer to 100%, thus validating the high-probability exact recovery statement.
- Please assume boundedness of the product $M\_\\Sigma M\_{\\Gamma,\\Gamma^T}$, where $M\_\\Sigma = \\max( \\| \\Sigma^{(1)} \\|\_{1,\\infty} , \\| \\Sigma^{(2)} \\|\_{1,\\infty} )$ and $M\_{\\Gamma,\\Gamma^T} = \\max( \\| \\Gamma\_{S,S}^{-1} \\|\_{1,\\infty} , \\| (\\Gamma\_{S,S}^T)^{-1} \\|\_{1,\\infty} )$, where $S$ is the support of the precision matrix $(\\Sigma^{(2)})^{-1} - (\\Sigma^{(1)})^{-1}$ and $\\Gamma = \\Sigma^{(2)} \\otimes \\Sigma^{(1)}$. I agree with the authors that this change is possible.
- Please check for typos (e.g., Line 250 and Algorithm 2 uses $\\Sigma_1,\\Sigma_2$ instead of $\\Sigma^{(1)},\\Sigma^{(2)}$.)
- Please take into account the comments from the reviewers, e.g., regarding additional literature comparison, exponential complexity.